# Research on Hazardous Waste Removal Management: Identification of the Hazardous Characteristics of Fluid Catalytic Cracking Spent Catalysts

**DOI:** 10.3390/molecules26082289

**Published:** 2021-04-15

**Authors:** Haihui Fu, Yan Chen, Tingting Liu, Xuemei Zhu, Yufei Yang, Haitao Song

**Affiliations:** 1Research Institute of Solid Waste, Chinese Research Academy of Environment Sciences, Beijing 100012, China; Fu_haihui123@126.com (H.F.); liutt@craes.org.cn (T.L.); zhuxm@craes.org.cn (X.Z.); 2State Environmental Protection Key Laboratory of Hazardous Waste Identification and Risk Control, Beijing 100012, China; 3Sinopec Research Institute of Petroleum Processing, Beijing 100083, China; cy.ripp@sinopec.com (Y.C.); songht.ripp@sinopec.com (H.S.)

**Keywords:** catalytic cracking unit, spent catalyst, hazardous characteristics, heavy metals, exemption management

## Abstract

Fluid catalytic cracking (FCC) spent catalysts are the most common catalysts produced by the petroleum refining industry in China. The National Hazardous Waste List (2016 edition) lists FCC spent catalysts as hazardous waste, but this listing is very controversial in the petroleum refining industry. This study collects samples of waste catalysts from seven domestic catalytic cracking units without antimony-based passivation agents and identifies their hazardous characteristics. FCC spent catalysts do not have the characteristics of flammability, corrosiveness, reactivity, or infectivity. Based on our analysis of the components and production process of the FCC spent catalysts, we focused on the hazardous characteristic of toxicity. Our results show that the leaching toxicity of the heavy metal pollutants nickel, copper, lead, and zinc in the FCC spent catalyst samples did not exceed the hazardous waste identification standards. Assuming that the standards for antimony and vanadium leachate are 100 times higher than that of the surface water and groundwater environmental quality standards, the leaching concentration of antimony and vanadium in the FCC spent catalyst of the G set of installations exceeds the standard, which may affect the environmental quality of surface water or groundwater. The quantities of toxic substances in all spent FCC catalysts, except those from G2, does not exceed the standard. The acute toxicity of FCC spent catalysts in all installations does not exceed the standard. Therefore, we exclude “waste catalysts from catalytic cracking units without antimony-based passivating agent passivation nickel agent” from the “National Hazardous Waste List.”

## 1. Introduction

Fluid catalytic cracking (FCC) is one of the major secondary operations for refining crude oil. FCC catalysts are widely used in the conversion of heavy feedstocks into lighter, more valuable products such as liquefied petroleum gases (LPG), cracked naphtha, and diesel oil [1,2,3]. In China, about 70% of gasoline and 33% of diesel are produced using this process [4,5]. As domestic raw materials become increasingly heavy and inferior, the amount of FCC spent catalysts produced by heavy metal deposition, wear, and hydrothermal deactivation is rising [6,7].

FCC spent catalysts make up the largest number of spent catalysts produced in the domestic petroleum refining industry, accounting for about 70% of the total annual spent catalyst production [8]. In 2016, “Spent Catalysts for Catalytic Cracking of Petroleum Products” was included in the “National Hazardous Waste List” (waste code 251-117-50), with toxicity listed as the hazardous characteristic. Pollutants in spent FCC catalysts come mainly from the catalytic cracking of feedstock oil and additives. There are many types of catalytic cracking feedstock oils, including residual oil, solvent deasphalted oil, and hydrotreated heavy oil. The characteristics of the feedstock added to different units vary, which results in large differences in the concentration of pollutants in the feedstock oil for catalytic cracking. The types and quantities of additives used in the production of complicated feedstocks also vary. With respect to the management of FCC spent catalysts in other countries, the United States has not included FCC spent catalysts in their hazardous waste list [9] and does not manage them as hazardous waste. In the “European Solid Waste/Hazardous Waste List,” FCC spent catalysts are classified as general solid wastes (waste number 160804). That document also states that “waste catalysts polluted by hazardous substances are not included,” which means that the EU manages only some FCC waste catalysts with excessive toxic content as hazardous waste. Therefore, whether it is reasonable to manage all the spent catalysts of catalytic cracking of petroleum products as hazardous wastes in China is a major dispute in the petroleum refining industry.

Currently, there are many studies on FCC spent catalysts, but most of them focus on the comprehensive utilization of FCC spent catalysts, such as extracting metals [1,10,11,12], adsorbent material [13,14], or cement raw material [15,16,17,18]. In terms of the hazardous characteristics of FCC spent catalysts, the main focus is on the morphology of the heavy metals nickel and vanadium. It is believed that vanadium in FCC spent catalysts exists in two valence states: V^5+^ and V^4+^. The main forms are vanadium pentoxide, vanadic acid, and sodium vanadate. Low-valent vanadium has not been detected [19,20,21]. Nickel in the spent FCC catalyst exists as Ni_x_Al_2_O_3+x_ (x ≤ 0.25) with a spinel-like structure. No NiO has been detected, indicating that there is no nickel oxide in the spent FCC catalyst or that the content of nickel oxide is much lower than 1000 mg/kg [22,23]. In terms of risk assessment, Bin (2019) studied the release of nickel, vanadium, and antimony in FCC spent catalysts under the most unfavorable environmental condition, which would affect the environmental quality of groundwater. Liu (2016) found that when FCC spent catalysts were directly landfilled, the leaching of the heavy metals nickel, zinc, barium, and arsenic posed certain risks to the environmental quality of groundwater. According to previous research [24,25], the leaching of some heavy metals in FCC spent catalysts poses certain risks to the environmental quality of groundwater.

According to preliminary research findings [24], the heavy metal content and leaching concentration of the two waste catalyst samples collected from the wax oil catalytic cracking unit are lower than the hazardous waste identification standard limit. The raw materials of the wax oil catalytic cracking unit are mostly straight-run wax oil, hydrogenated wax oil, hydrogenated heavy oil, and hydrogenated wax oil. After the feedstock oil is catalytically hydrogenated, most of the heavy metals can be removed [26]. In order to facilitate the management of FCC spent catalysts, this study investigates the hazardous characteristics of spent catalysts produced in catalytic cracking units without antimony-based passive nickel agent in the production process. This research will support the scientific classification and grading management of FCC spent catalysts and also provide technical support for the revision of the HW 251-017-50 category of waste in the National Hazardous Waste List.

## 2. Results and Discussion

### 2.1. Screening of Characteristic Pollutants of FCC Spent Catalysts

#### 2.1.1. Analysis of the Hazardous Characteristics of FCC Spent Catalysts

In terms of mineralogy, FCC spent catalysts are faujasite, a type of zeolite. The main components are alumina and silica, which account for about 95% of the total. The spent catalyst is discharged after oxygen enrichment regeneration over a temperature range of 680–700 °C. Therefore, FCC spent catalysts do not possess the characteristics of flammability and reactivity. The spent FCC catalyst is weakly acidic or neutral, without the characteristics of corrosive danger. Therefore, this research focuses on the leaching toxicity, toxic substance content, and acute toxicity hazard characteristics of spent FCC catalysts.

#### 2.1.2. Screening of Characteristic Organic Pollutants

According to the FCC spent catalyst production process, there is a small amount of carbon deposits in FCC spent catalysts. The results of seven samples selected from catalytic cracked units (A–G) are shown in Table 1.

The test results show that the carbon content in the FCC spent catalyst samples is low. Therefore, there may be a small number of organic pollutants in the FCC waste catalyst. Seven samples were selected from each of seven FCC units to test for 122 kinds of volatile and semivolatile organic pollutants. The results are shown in Figure 1.

Only 14 kinds of organic compounds were detected. The concentrations of organic matter in the FCC spent catalyst are much lower than those of the standard limit of “Identification of Hazardous Wastes Identification Standard for Leaching Toxicity” (GB 5085.3-2007) and “Identification of Toxic Substances Content of Hazardous Waste Identification Standard” (GB 5085.6-2007). Since the wax oil catalytic cracking unit uses petroleum distillate as raw material, petroleum solvent is used as the characteristic pollutant for follow-up research.

#### 2.1.3. Screening of Characteristic Heavy Metal Pollutants

The heavy metals in FCC spent catalysts come mainly from catalytic cracking feedstock oil, FCC catalysts, and auxiliary agents. The test results of the main heavy metals in catalytic cracking feedstock oil are shown in Table 2. The main/co-catalysts and main components added in the production process are shown in Table 3.

Nickel, vanadium, antimony, copper, cobalt, and zinc were chosen as characteristic heavy metal pollutants. The selection of these elements is supported by analysis of the main heavy metals in feedstock oil, the main components of the main/promoter catalysts, and previous research on the characteristics of heavy metals in FCC spent catalysts [24].

### 2.2. Study on the Content of Toxic Substances in FCC Spent Catalysts

#### 2.2.1. Heavy Metal Content in FCC Waste Catalysts in Different Units

A total of 21 FCC spent catalyst samples from seven sets of catalytic cracking units without antimony-based passivators were tested for heavy metals and petroleum hydrocarbons. The results are shown in Figure 2.

Figure 2 shows that the concentrations of characteristic pollutants in FCC spent catalysts collected from different catalytic cracking units are different. By analyzing the overall concentration of heavy metals, it can be seen that the concentrations of heavy metals such as vanadium and nickel are relatively high, while those of copper, cobalt, and zinc are relatively low. This trend can be attributed to differences in the raw materials of different units. The different raw materials are highly correlated with the different concentrations of characteristic heavy metals [27,28].

The concentrations of petroleum hydrocarbons in the FCC waste catalysts collected from the F and G units are relatively low, both below 250 mg/kg. However, the concentrations of the petroleum hydrocarbons of other FCC spent catalysts are all between 400 and 650 mg/kg. The reason for the low concentrations in samples from the F and G units is that in these units, the spent FCC catalyst has to go through a high-temperature coking regeneration stage before entering the waste agent tank. The petroleum hydrocarbon adhering to the catalyst surface is basically removed during the regeneration process, and thus, the concentration of petroleum hydrocarbon is relatively low.

Considering the heavy metal concentrations in the waste FCC catalyst of a single unit, the contents of nickel and vanadium in the waste FCC catalysts of the F and G units are relatively high, both above 250 mg/kg. Only antimony was detected in the FCC waste catalyst of the G unit, at a concentration above 400 mg/kg. This may be related to the fact that the G unit had undergone shutdown and overhaul within half a year before sampling. A large amount of waste catalyst containing antimony-based passive nickel agent produced by non-wax oil catalytic cracking unit was used as the starting agent when it started, resulting in high antimony concentration in the spent catalyst.

#### 2.2.2. Calculation of Toxic Substance Content

To identify the inorganic toxic substances content, the heavy metal content must be converted into the content of inorganic toxic compounds containing heavy metals. In the process of calculating the content of toxic substances, a worst-case scenario is adopted to screen compounds for heavy metals. In other words, if the compound contained the same type of heavy metal, but the type of compound could not be determined, the compound with the largest molecular weight and the lowest identification standard value was selected. Since the nickel content in the FCC spent catalyst sample was relatively low, nickel was calculated as nickel dioxide. The toxic substance “V” is elemental vanadium, which is not considered in this calculation (Table 4). It should be noted that a compound being selected for calculation does not necessarily mean the compound was present in the waste.

The average value of each sample from the different units was used to calculate the cumulative toxicity. The results are shown in Figure 3. The results show that the content of highly toxic substances and toxic substances in all samples did not exceed the corresponding limits of 0.1% and 3% in the “Identification of Toxic Substances in the Identification Standard for Hazardous Wastes” (GB 5085.6-2007). Except for the carcinogenic substance content and cumulative toxicity of the G2 sample exceeding the limit of 0.1% and 1 in the GB 5085.6-2007, the other samples did not exceed the standard limit. By analyzing the carcinogenic substances and cumulative toxicity of sample G2, we found that the toxicity comes mainly from nickel dioxide. The content of a single nickel dioxide substance exceeded the 0.1% limit for carcinogenic substances. As reported in studies of the nickel form in FCC spent catalysts, nickel exists mainly in the form of a spinel-like structure NixAl_2_O_3+x_ (x ≤ 0.25). A reverse calculation of toxicity based on the 0.1% limit of carcinogenic substance content finds that when the content of nickel in the form of a spinel-like structure comprises more than 11% of the total nickel content, the content of toxic substances from G2 does not exceed the standard. Therefore, the possibility that G2 has the hazardous characteristics of a toxic substance is extremely small.

### 2.3. Study on the Toxicity of FCC Spent Catalysts

A total of 21 FCC spent catalyst samples from seven sets of catalytic cracking units without antimony-based passivators were tested for heavy metal leaching. The results are shown in Figure 4.

Figure 4 shows that the detection rates of nickel and vanadium in the spent catalyst samples are relatively high, while the detection rates of Cu, Co, and Zn are relatively low. The leaching concentrations are also very low, all below 0.2 mg/L. From the perspective of a single set of equipment, the leaching concentrations of nickel, vanadium, and antimony in the waste catalyst samples collected by the G set of equipment are higher than those of other FCC waste catalyst samples, which is consistent with the high content of nickel, vanadium, and antimony in the FCC waste catalyst collected from the G set of equipment.

The leaching concentrations of nickel, copper, and zinc in all FCC spent catalyst samples are below the limits in the “Leaching Toxicity of Hazardous Waste Identification Standard” (GB5085.3-2007). Unfortunately, there are no standards for the leaching limit of vanadium, antimony, and cobalt. Therefore, we used the Class III standard limits from “Groundwater Quality Standards” (GB/T14848-2017) and the specific project standard limits for centralized drinking water sources from “Surface Water Environmental Quality Standards,” applied a dilution attenuation factor (DAF) of 100, and calculated standards of 5 mg/L for vanadium, 0.5 mg/L for antimony, and 5 mg/L for cobalt. The leaching concentration of cobalt in all samples is lower than 5 mg/L, but antimony and vanadium in waste catalyst samples from the G unit exceed the standards. These results indicate that if this FCC waste catalyst is disposed of in an irregular landfill or ground storage, the metal antimony and vanadium may leach out in acid rain, which can further affect the environmental quality of surface water or groundwater.

In summary, except for the samples from the G unit, samples of the FCC spent catalyst do not have the characteristics of a leaching toxicity hazard.

### 2.4. Acute Toxicity Study of FCC Spent Catalyst

The acute toxicity was estimated by considering the waste catalyst sample to be a mixture of the detected toxic substances and using data from the test results of the toxic substance content of the samples and the “Chemical Classification and Labeling Specification Part 18: Acute Toxicity” (GB 30000.18). The main components of the catalyst are alumina and silica (their combined content is above 90%) [27,29]; hence, the calculation and caveats are as follows:(1)$$\mathrm{ATE}=\frac{100}{\sum \frac{\mathrm{Ci}}{\mathrm{ATEi}}}$$
where ATE is the estimated value of the acute toxicity of solid waste, Ci is the percentage of the i-th toxic substance contained in the solid waste, and ATEi is the acute toxicity data of the i-th toxic substance.

When calculating acute toxicity using this formula, the acute toxicity of substances such as water and sugar are ignored, as are the acute toxicities of substances with LD50 greater than 2000 mg/kg. The main substances in the waste catalyst are silica, which has an LD50 of 22,500 mg/kg, and alumina, which has an LD50 of >3600 mg/kg. This substance is ignored in the acute toxicity calculation process. In addition, although vanadium pentoxide is not included in the “Identification of Toxic Substances in the Identification Standards for Hazardous Wastes” (GB 5085.6-2007), it was included in the acute toxicity calculation because vanadium pentoxide is relatively toxic (oral LD50 for mice is 5 mg/kg).

The result of the parameter estimation in Table 4 and Table 5 is shown in Figure 5. The oral acute toxicity of the FCC spent catalyst is calculated to exceed 1174 mg/kg, which is far greater than the limit (≤200 mg/kg) specified in the “Acute Toxicity Screening Standard for Hazardous Waste Identification” (GB 5085.2-2007). Therefore, the FCC spent catalyst is not acutely toxic.

## 3. Materials and Methods

### 3.1. Materials

Using the “Compilation of Basic Data of Refining Production Plants in 2018” as a reference, a total of seven FCC units without antimony-based nickel passivating agent were selected from 49 Sinopec refineries based on the composition of the feedstock oil and the heavy metal content in FCC waste catalysts. Basic characteristics are shown in Table 6. From 15 to 23 August 2019, seven sets of FCC waste catalyst samples were collected. A total of 21 waste catalyst samples were collected at three different times.

### 3.2. Experimental Methods

#### 3.2.1. Heavy Metal Test Methods

We used an FCC spent catalyst leaching toxicity test based on the “Identification Standard for Hazardous Waste: Identification of Leaching Toxicity” (GB5085.3-2007). The “Solid Waste Leaching Toxicity Leaching Method-Sulfuric Acid and Nitric Acid Method” (HJ/T299-2007) was used to prepare the leachate, using a liquid–solid ratio of 10:1.

Heavy metals test in FCC spent catalysts is based on “Determination of 22 Metal Elements in Solid Waste-Inductively Coupled Plasma Emission Spectrometry” (HJ 781-2016).

Heavy metals test in feedstock oil is based on “Simultaneous Determination of 14 Trace Elements in Crude Oil and Heavy Oil by Inductively Coupled Plasma Atomic Emission Spectrometry (ICP-AES)” (RIPP124-90).

#### 3.2.2. Organic Test Methods

The volatile organic compound test was based on “Determination of Volatile Organic Compounds in Soils and Sediments Purge and Trap-Gas Chromatography–Mass Spectrometry” (HJ 605-2011). A total of 58 volatile organic pollutants in FCC spent catalysts were tested. The organic matter was extracted and prepared according to the “Pressurized Fluid Extraction Method for the Extraction of Solid Waste Organic Matter” (HJ 782-2016). A total of 64 semivolatile organic pollutants in FCC spent catalysts were tested according to the “Determination of Semivolatile Organic Compound of Solid Waste by Gas Chromatography–Mass Spectrometry” (HJ 951-2018).

Petroleum hydrocarbons were tested in accordance with “Identification Standards for Hazardous Wastes-Identification of Toxic Substances Content” (Appendix O: Determination of Total Recyclable Petroleum Hydrocarbons in Solid Wastes by Infrared Spectroscopy) (GB5085.6-2007).

#### 3.2.3. Carbon Test Method

Content of carbon deposition in FCC spent catalyst was determined by using a carbon/sulfur analyzer (LECO CS844), based on “Determination of Carbon and Sulfur in Catalysts Produced by Petroleum Refining High-Frequency Combustion Infrared Absorption Method” (HG/T 5594).

## 4. Conclusions

Based on the standards in the “Hazardous Waste Identification Standard” (GB 5085.1~7-2007), our study on the hazardous characteristics of waste catalysts produced by FCC units without antimony-based passivators concluded that FCC waste catalysts are not flammable, corrosive, or reactive. Furthermore, our leaching toxicity and toxic substance study found that, except for the antimony and vanadium in the FCC waste catalyst collected by the G set of equipment, the waste catalyst samples collected do have leaching toxicity or toxic substance content hazard characteristics. Finally, our acute toxicity study found that all FCC spent catalyst samples do not have acute toxicity hazard characteristics.

Domestic FCC catalysts are mainly aluminum-based catalysts (the binder is alumina), which have a considerable degree of passivation nickel space. According to industry experts, the nickel content of feedstock oil within 6 ppm can be considered without an antimony-based passivation agent. Based on these research results on the hazardous characteristics of waste catalysts produced by seven sets of catalytic cracking units without antimony-based passivators, we recommended that “waste catalysts produced by catalytic cracking units without antimony-based passivators” be excluded from the “National Hazardous Waste List.”

In order for this kind of waste catalyst to be excluded, an antimony-based deactivator cannot be added to the balancer reaction process and regeneration process of the unit. Furthermore, when the unit needs to use a low-activity balancer during operation or after maintenance, waste catalysts from an FCC unit with an antimony passivator should not be used. If these waste catalysts are used, it is necessary to determine whether the waste catalysts produced by the unit are hazardous wastes and if the waste catalysts identified as hazardous wastes are being managed as hazardous wastes.

## Figures and Tables

**Figure 1 molecules-26-02289-f001:**
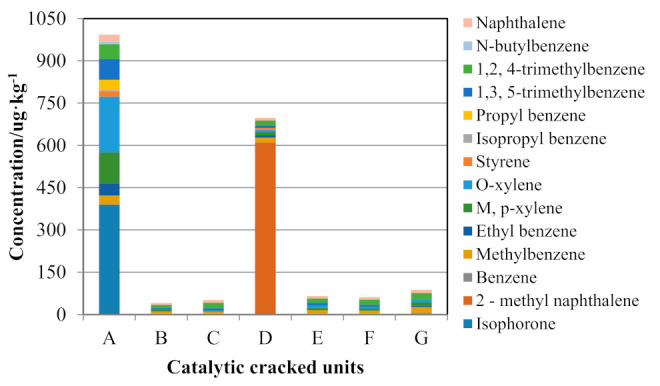
Organic matter concentrations in the spent catalyst.

**Figure 2 molecules-26-02289-f002:**
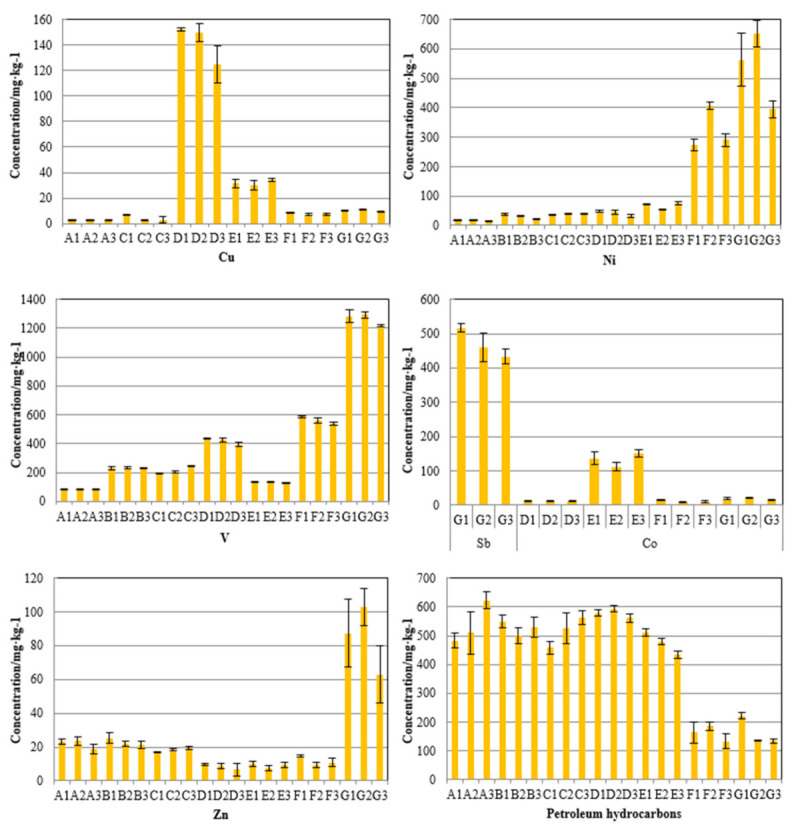
FCC spent catalyst characteristic pollutant concentration.

**Figure 3 molecules-26-02289-f003:**
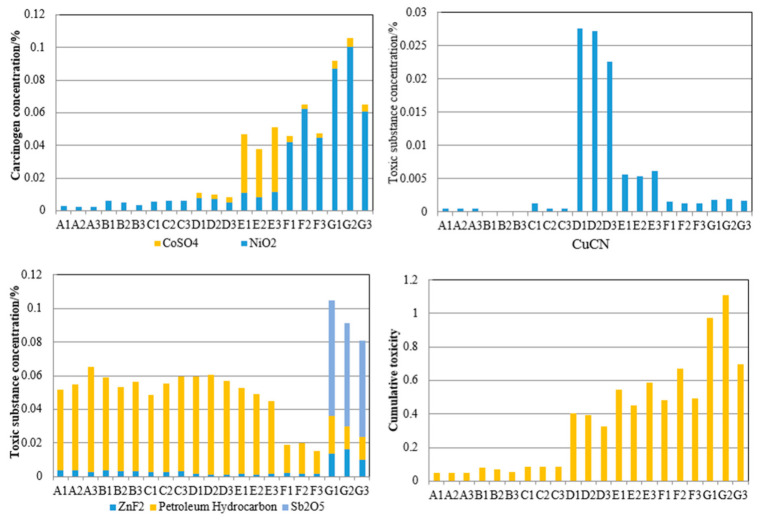
Cumulative toxicity of toxic substances in FCC spent catalyst samples from different units.

**Figure 4 molecules-26-02289-f004:**
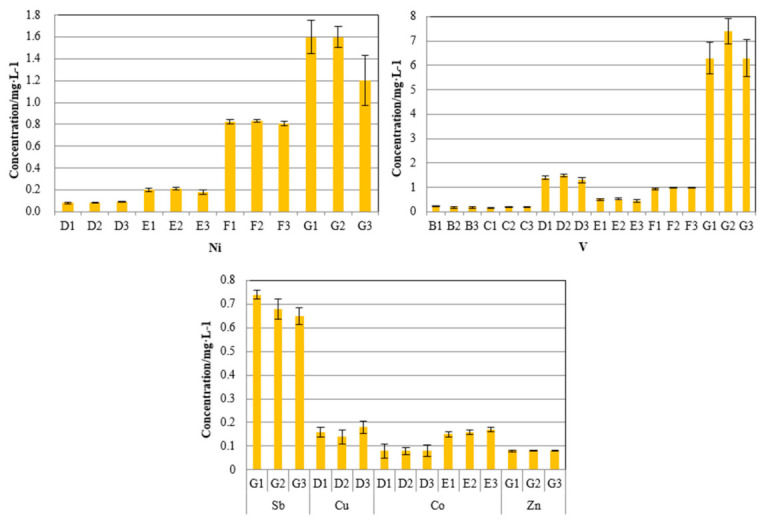
Heavy metal leaching concentration of spent catalysts.

**Figure 5 molecules-26-02289-f005:**
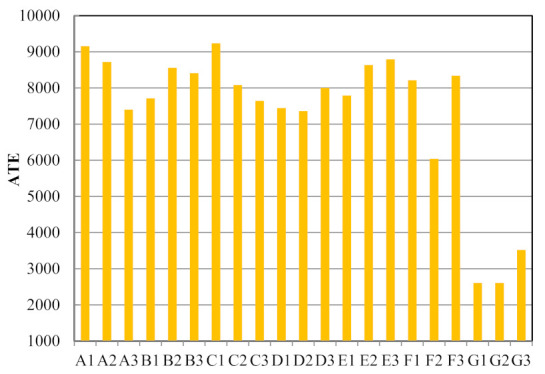
Acute toxicity estimates of FCC spent catalyst samples from different units.

**Table 1 molecules-26-02289-t001:** Carbon content in FCC spent catalysts.

Set	A	B	C	D	E	F	G
**C/wt.%**	0.028	0.028	0.016	0.032	0.020	0.018	0.050

**Table 2 molecules-26-02289-t002:** Heavy metals concentrations in catalytic cracking feedstock oil.

Set	Fe	Ni	V	Na	Ca	Cu
A	0.98	0.14	0.21	0.95	0.97	0.06
B	1.85	0.16	0.17	1.04	1.21	0.17
C	0.75	0.19	0.96	0.22	0.42	0.07
D	1.20	0.7	0.99	0.13	0.23	0.08
E	1.36	0.69	0.57	0.71	1.38	0.10
F	2.90	2.72	1.80	0.42	2.77	ND
G	5.51	4.24	2.27	0.95	3.95	0.02

**Table 3 molecules-26-02289-t003:** Main/co-catalyst characteristic pollutants.

Main/Auxiliary	Main Ingredients
FCC catalyst	Al_2_O_3_, SiO_2_, Na_2_O, SO_4_^2−^, Fe_2_O_3_, etc.
CO combustion denitrifier	Al_2_O_3_, Fe_2_O_3_, Pd, Pt, and other precious metals and rare earth oxides
CO combustion aid	Support Al_2_O_3_ or SiO_2_–Al_2_O_3_, active components platinum, palladium, and other heavy metals, Na_2_O
Sulfur transfer agent	Al_2_O_3_, MgO, La_2_O_3_, V_2_O_5_, etc.
Octane additive	SiO_2_, Al_2_O_3_, trace rare earth elements, trace Na_2_O, Fe, SO_4_^2^^−^, and Cl^−^

**Table 4 molecules-26-02289-t004:** Selection of compounds for calculation of inorganic toxic substance content of the spent catalyst.

Pollutants	Corresponding Compound	Toxicity Category	Conversion Factor
Co	CoSO_4_	Carcinogen	155/59
Ni	NiO_2_	Carcinogen	91/59
Cu	CuCN	Highly toxic substance	89.5/64
Zn	ZnF_2_	Toxic Chemical	103/65
Sb	Sb_2_O_5_	Toxic Chemical	323.5/243.5
Petroleum Hydrocarbon	Petroleum Hydrocarbon	Toxic Chemical	1

**Table 5 molecules-26-02289-t005:** Acute toxicity estimation parameters. Unit: mg/kg.

Toxic Substances	CoSO_4_	Pb_3_(PO_4_)_2_	CuCN	ZnF_2_ ^2)^	V_2_O_5_	Sb_2_O_5_ ^2)^	NiO_2_ ^2)^	Petroleum Hydrocarbon ^2)^
Oral LD50 ^1)^	389	540	500	5	5	5	5	5

^1)^ Acute toxicity parameters are from the Material Safety Data Sheet (MSDS) database of hazardous chemicals; ^2)^ “Chemical Classification and Labeling Specifications Part 18: Acute Toxicity” (GB 30000.18) category 1 limit calculation.

**Table 6 molecules-26-02289-t006:** Characteristics of catalytic cracking units sampled.

Set	Type and Proportion of Feedstocks	Information of Feedstocks				Types of Additives	Capacity (10 K t/year)	Regenerator Temperature (°C)
		Density (20 °C) (kg/m^3^)	S (%)	Initial Distillation Point (°C)	Final Distillation Point (°C)			
A	Hydrogenated diesel 100%	910	0.02	170	360	No addition	69	680
B	Hydrogenated straight-run wax oil 55%, hydrodeasphalted oil 35%, purchased wax oil 10%	909	0.24	219	737	CO combustion denitrification agent, sulfur transfer agent, octane booster	230	690
C	Hydrogenated wax oil 100%	895	0.41	211	556	CO combustion promoter, sulfur transfer agent	290	680
D	Straight-run wax oil 50%, hydrogenated wax oil 32%, catalytic feedstock oil 14%, naphtha 3%, etc.	917	1.44	221	599	CO combustion denitrifier, CO combustion promoter, octane booster	69	680–700
E	Hydrogenated wax oil 100%	899	0.06	284	508	CO combustion denitration agent, sulfur transfer agent	65	680
F	Hydrogenated wax oil 51%, hydrogenated residue 30%, hydrogenated diesel 13%, 3% vacuum gas oil, etc.	955	0.19	218	566	CO combustion aid	260	680
G	Wax oil 35%, hydrogenated heavy oil (including hydrogenated residue about 40%) 40%, hydrogenated diesel 25%	921	0.53	243	574	CO combustion aid	120	695

## Data Availability

Not applicable.

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
