# Peer review of "Research on Hazardous Waste Removal Management: Identification of the Hazardous Characteristics of Fluid Catalytic Cracking Spent Catalysts"

_molecules, 2021, doi:10.3390/molecules26082289_

Round 1
Reviewer 1 Report
Title: Research on hazardous waste removal management: identification of the hazardous characteristics of fluid catalytic cracking spent catalysts
Recommendation: Manuscript should be published after major revision.
Comments
This paper is focusing on the very important matter of the hazardous waste removal management of spent FCC catalysts. The contaminant metal levels of the samples in the present study are considered as relatively low to moderate for the FCC community, because they are coming from units that are processing relatively light and mainly hydrotreated feed. The phrase «waste catalysts produced by catalytic cracking units without antimony-based passivators» is not so suitable for the samples under study. The G set of samples contains antimony even though it is a result of contamination with another catalyst. Moreover, there are spent catalysts from Sb-free FCC units with higher metal levels than the samples of this study. Therefore, the range of each metal’s concentration is not wide enough in order to support a statement about samples «produced by catalytic cracking units without antimony-based passivators». The authors conclude that «waste catalysts produced by catalytic cracking units without antimony-based passivators» should be excluded from the National Hazardous Waste List, but this recommendation is very generic, as explained earlier, and not actually reflecting the findings of the study. The problem is that the leaching toxicity (V and Sb) of the G set of samples exceeds the standards as claimed by the authors in the 4th and 5th line of page 9: «.. except for the samples from the G device, samples of the FCC spent catalyst do not have the characteristics of a leaching toxicity hazard. » The authors should investigate the V and Sb concentration limits resulting in acceptable toxicity and leaching toxicity. Probably the same analysis should be performed for the rest contaminant metals. Then the recommendation could be more specific about waste catalytic cracking catalysts with contaminant metals (Ni, V, Sb, Zn, Cu, Co) below certain concentrations. Finally, the language of the manuscript needs to be improved.
Specific Comments
- In Figure 2 the size of the bar diagrams is not uniform. Also, the X axes are preferred to be similar to the Sb and Co bar diagram so that every bar is labeled. The same amendment is recommended for the X axis in Figure 5.
Author Response
Point 1: The contaminant metal levels of the samples in the present study are considered as relatively low to moderate for the FCC community, because they are coming from units that are processing relatively light and mainly hydrotreated feed. The phrase «waste catalysts produced by catalytic cracking units without antimony-based passivators» is not so suitable for the samples under study. The G set of samples contains antimony even though it is a result of contamination with another catalyst. Moreover, there are spent catalysts from Sb-free FCC units with higher metal levels than the samples of this study. Therefore, the range of each metal’s concentration is not wide enough in order to support a statement about samples «produced by catalytic cracking units without antimony-based passivators». The authors conclude that «waste catalysts produced by catalytic cracking units without antimony-based passivators» should be excluded from the National Hazardous Waste List, but this recommendation is very generic, as explained earlier, and not actually reflecting the findings of the study. The problem is that the leaching toxicity (V and Sb) of the G set of samples exceeds the standards as claimed by the authors in the 4th and 5th line of page 9: «.. except for the samples from the G device, samples of the FCC spent catalyst do not have the characteristics of a leaching toxicity hazard. » The authors should investigate the V and Sb concentration limits resulting in acceptable toxicity and leaching toxicity. Probably the same analysis should be performed for the rest contaminant metals. Then the recommendation could be more specific about waste catalytic cracking catalysts with contaminant metals (Ni, V, Sb, Zn, Cu, Co) below certain concentrations. Finally, the language of the manuscript needs to be improved.
Response 1: In China, wastes listed in the " National Hazardous Waste List " must be managed as hazardous wastes. FCC spent catalysts were listed in the “National Hazardous Waste List” in 2016. Refining companies have a lot of controversy about this. Therefore, we conducted research on the hazardous characteristics of FCC spent catalysts and found that the environmental risks of FCC spent catalysts mainly come from heavy metals Ni, V, Sb, when the nickel content in the feedstock oil is high, an antimony-based passivator needs to be added to extend the service life of the catalyst. The antimony-based passivator is the main source of Sb in the spent catalyst. In addition, in order to facilitate the local management, we consider first to study the catalytic cracking unit without antimony-based passivator in the production process. Therefore, the equipment without antimony-based passivator in the Sinopec company was sampled and studied, and it was found that except for the leaching concentration of V and Sb in the samples of the G device, which exceeds the standard value, the spent catalysts produced by the other devices are not hazardous wastes. Therefore, this part of the spent catalysts can be excluded from the National Hazardous Waste List.
During the operation of the G device, the waste catalyst produced by the catalytic cracking unit using the antimony-based passivator is added. Therefore, it is necessary to determine whether the waste catalysts produced by the device are hazardous wastes, and if the waste catalysts identified as hazardous wastes are being managed as hazardous wastes. Therefore, in part 4 of the manuscript, we recommend that catalytic cracking units that do not use antimony-based passivators do not use spent catalysts from catalytic cracking units that add antimony-based passivators during operation.
The suggestion for conducting research on spent catalytic cracking catalysts with a concentration of heavy metal pollutants below a certain level is very good. On the basis of our existing research, we will conduct follow-up research, thank you very much!
Point 2: In Figure 2 the size of the bar diagrams is not uniform. Also, the X axes are preferred to be similar to the Sb and Co bar diagram so that every bar is labeled. The same amendment is recommended for the X axis in Figure 5.
Response 2: All the figures in the manuscript have been adjusted to a unified format.

Reviewer 2 Report
It is an interesting paper in which a study of the hazardous characteristics of fluid catalytic cracking spent catalysts is done.
The paper is well written, and the conclusions seem logical but there are some information gaps that are essential to deepen the conclusions of the paper. The paper merits publication once the authors respond to the following items:
- Table 1 shows the characteristics of the sampled FCC units, but the information on the type of feedstock (hydrogenated diesel 100%, hydrogenated wax oil 100%, and so on) is insufficient: it is a description that is subject to interpretation.The authors should indicate some basic properties of the feedstocks such as initial and final points of the distillation curve, density, sulfur content and, above all, metals content (Ni, V, Fe, ...).
- In the catalyst samples obtained, it is understood that the catalysts were removed from the regenerator, but the coke content in that regenerated catalyst is not indicated. The catalysts are not totally regenerated, that is, they do not have 0.0% coke content, and it can vary from one FCC unit to another.This can especially affect the content of organic pollulants when comparing the catalysts from the different units.The authors should include a table with all the sampled catalysts and their coke content.
- Concerning the catalysts, it is not clear if the ppm of metallic pollulants comes from the composition of the catalysts themselves or from the accumulation of those metals after use in the FCC unit.In the paper, the comparison of the concentrations of metallic pollulants obtained from fresh catalysts is missing.
I think without that information, the results of the article are not sound enough.
Author Response
- Point 1: Table 1 shows the characteristics of the sampled FCC units, but the information on the type of feedstock (hydrogenated diesel 100%, hydrogenated wax oil 100%, and so on) is insufficient: it is a description that is subject to interpretation.The authors should indicate some basic properties of the feedstocks such as initial and final points of the distillation curve, density, sulfur content and, above all, metals content (Ni, V, Fe, ...).
Response 1: The test data for the concentration of heavy metals in the feedstock oil has been added in Section 3.1(3), that is, lines 166-167. At the same time, the test method for heavy metals in the feedstock oil is added in Section 2.2.1, that is, lines 113-115.
Point 2: In the catalyst samples obtained, it is understood that the catalysts were removed from the regenerator, but the coke content in that regenerated catalyst is not indicated. The catalysts are not totally regenerated, that is, they do not have 0.0% coke content, and it can vary from one FCC unit to another.This can especially affect the content of organic pollulants when comparing the catalysts from the different units.The authors should include a table with all the sampled catalysts and their coke content.
Response 2:The test data of carbon content in spent FCC catalysts has been added to section 3.1(2), which is lines 145-149, and the carbon content test method is added to section 2.2.3, which is lines 129-133.
- Point 3: Concerning the catalysts, it is not clear if the ppm of metallic pollulants comes from the composition of the catalysts themselves or from the accumulation of those metals after use in the FCC unit.In the paper, the comparison of the concentrations of metallic pollulants obtained from fresh catalysts is missing.
Response 3: The main components of the fresh FCC catalyst are shown in Table 4. The main components of the new FCC catalyst include Al2O3, SiO2, Na2O, SO42-, Fe2O3 and a small amount of rare earth metals. In addition, our previous research has conducted a full element analysis of the heavy metals in the spent catalyst (( Bin et al., 2019)), therefore, without testing the concentration of heavy metals in the fresh FCC catalyst, the types of characteristic heavy metals in the spent FCC catalyst can also be determined, so the concentration of heavy metals in the fresh FCC catalyst has not been tested.

Round 2
Reviewer 1 Report
Review of manuscript V.2
Authors: Wei Shumei, Xu Yarong, Zhu Xuedong
Title: Research on hazardous waste removal management: identification of the hazardous characteristics of fluid catalytic cracking spent catalysts
Recommendation: Manuscript should be published after revision.
Comments
It is clear from the paper and the authors response that waste catalysts from FCC units utilizing Sb based Ni passivators cannot be excluded from the " National Hazardous Waste List ". On the other hand, the authors did not make any comment about the other contaminant metals concentration range. Are the metal levels of the samples of this study the highest among Sb-free units in China? Are these 7 FCC units the only Sb-free units among all Sinopec FCC units? The authors should claim whether the metal concentrations of the samples of this study are the highest when comparing all Sb-free FCC units in China. Could the authors provide information about the metal concentrations in all Sb-free FCC units in China (at least a concentration range without data from every unit)? Only after proving that you cannot get higher metal levels from Sb-free FCC units in China, the statement about exclusion of the waste catalysts from SB-free units from the National Hazardous Waste List is strong. In case there are waste FCC catalysts from Sb-free units in China containing higher metal concentrations than the samples of the present study, then the conclusion is only partially proved.
Specific Comments
- Paragraph 2.1 needs amendment. In the phrase “…a total of seven sets of FCC plants without antimony-based...” the word “sets” should be erased. So, it should be: “seven FCC plants without antimony-based...”. Also, the last sentences are confusing.: “From August 15 to 23, 2019, seven sets of FCC waste catalyst samples were collected. A total of 21 waste catalyst samples were collected at seven different times.” Probably the last sentence should be changed to: ”… were collected at three different times.” in order to count 21 samples from 7 FCC units.
- The word “device” is not appropriate for FCC units. Please rephrase throughout the paper.
Author Response
Point 1: The authors did not make any comment about the other contaminant metals concentration range. Are the metal levels of the samples of this study the highest among Sb-free units in China? Are these 7 FCC units the only Sb-free units among all Sinopec FCC units? The authors should claim whether the metal concentrations of the samples of this study are the highest when comparing all Sb-free FCC units in China. Could the authors provide information about the metal concentrations in all Sb-free FCC units in China (at least a concentration range without data from every unit)? Only after proving that you cannot get higher metal levels from Sb-free FCC units in China, the statement about exclusion of the waste catalysts from SB-free units from the National Hazardous Waste List is strong. In case there are waste FCC catalysts from Sb-free units in China containing higher metal concentrations than the samples of the present study, then the conclusion is only partially proved.
Response 1:This study is a follow-up study carried out on the basis of Bin et al (2019). In the paper, 3 samples were randomly selected for full analysis of 17 heavy metals, and only 10 heavy metals were detected, respectively, aluminum, vanadium, chromium, iron, cobalt, nickel, copper, zinc, antimony and lead. Among them, aluminum and iron have a low risk of being absorbed by the human body through exposure, while the concentrations of chromium, iron, copper, and lead are all less than 0.01%, and the concentration is relatively Low. Therefore, this study is based on the research of Bin et al (2019), combined with the main heavy metal types in the feedstocks and main ingredients of main/co-catalyst, Nickel, vanadium, antimony, copper, cobalt, and zinc were chosen as characteristic heavy metal pollutants in this study.
In addition to the 7 FCC units without antimony-based nickel passivation agent in this study, Sinopec also has 2 FCC units without antimony-based nickel passivation agent, and the data has been published in Bin et al (2019). this study is a follow-up study based on the discovery of the low concentration of characteristic pollutants in the spent catalysts produced by the two FCC units.
According to official data from PetroChina, Sinopec, and CNOOC, as well as regular inspection data from catalyst manufacturers, there are 18 FCC units in China without antimony-based passivators. This study includes units with different ranges of nickel and vanadium content in the feedstock, which is low (15 units), medium (3 units), high (1unit), that is, it already contains the unit with the highest heavy metal content in the spent catalyst. However, the highest concentration of heavy metal nickel in the samples collected in this study is relatively low. The highest concentration of heavy metal nickel in the waste catalyst produced by the FCC unit without antimony-based passive nickel agent using high metal content feedstock can also reach 2000ppm. The highest concentration of other heavy metals is basically the same as this study. According to foreign and our previous research (Bin et al. 2009), the nickel concentration is 2000ppm, considering that the nickel in the waste catalyst is mainly in nickel-aluminum spinel, which will not exceed the limit of hazardous waste identification standards.
Point 2: Specific Comments
- Paragraph 2.1 needs amendment. In the phrase “…a total of seven sets of FCC plants without antimony-based...” the word “sets” should be erased. So, it should be: “seven FCC plants without antimony-based...”. Also, the last sentences are confusing.: “From August 15 to 23, 2019, seven sets of FCC waste catalyst samples were collected. A total of 21 waste catalyst samples were collected at seven different times.” Probably the last sentence should be changed to: ”…were collected at three different times.” in order to count 21 samples from 7 FCC units.
Response 2: The phrase “…a total of seven sets of FCC plants without antimony-based...” has been changed to” …a total of seven FCC units without antimony-based...”.
The last sentence “A total of 21 waste catalyst samples were collected at seven different times.”has been changed to“A total of 21 waste catalyst samples were collected at three different times”.
- The word “device” is not appropriate for FCC units. Please rephrase throughout the paper.
Response 3: Replaced all "device" in the paper with "unit".
Reviewer 2 Report
The manuscript has been significantly improved, but the information on the type of FCCU feedstocks in Table 1 is missing. It is a minor matter but it would have improve the paper.
Author Response
Point 1: The manuscript has been significantly improved, but the information on the type of FCCU feedstocks in Table 1 is missing. It is a minor matter but it would have improve the paper.
Response 1: The information on the type of feedstocks is added in Table 6.